# Regulation of EGFR Endocytosis by CBL During Mitosis

**DOI:** 10.3390/cells7120257

**Published:** 2018-12-07

**Authors:** Ping Wee, Zhixiang Wang

**Affiliations:** Department of Medical Genetics and Signal Transduction Research Group, Faculty of Medicine and Dentistry, University of Alberta, Edmonton, AB T6G 2H7, Canada; pwee@ualberta.ca

**Keywords:** EGFR, mitosis, endocytosis, CBL, ubiquitination

## Abstract

The overactivation of epidermal growth factor (EGF) receptor (EGFR) is implicated in various cancers. Endocytosis plays an important role in EGFR-mediated cell signaling. We previously found that EGFR endocytosis during mitosis is mediated differently from interphase. While the regulation of EGFR endocytosis in interphase is well understood, little is known regarding the regulation of EGFR endocytosis during mitosis. Here, we found that contrary to interphase cells, mitotic EGFR endocytosis is more reliant on the activation of the E3 ligase CBL. By transfecting HeLa, MCF-7, and 293T cells with CBL siRNA or dominant-negative 70z-CBL, we found that at high EGF doses, CBL is required for EGFR endocytosis in mitotic cells, but not in interphase cells. In addition, the endocytosis of mutant EGFR Y1045F-YFP (mutation at the direct CBL binding site) is strongly delayed. The endocytosis of truncated EGFR Δ1044-YFP that does not bind to CBL is completely inhibited in mitosis. Moreover, EGF induces stronger ubiquitination of mitotic EGFR than interphase EGFR, and mitotic EGFR is trafficked to lysosomes for degradation. Furthermore, we showed that, different from interphase, low doses of EGF still stimulate EGFR endocytosis by non-clathrin mediated endocytosis (NCE) in mitosis. Contrary to interphase, CBL and the CBL-binding regions of EGFR are required for mitotic EGFR endocytosis at low doses. This is due to the mitotic ubiquitination of the EGFR even at low EGF doses. We conclude that mitotic EGFR endocytosis exclusively proceeds through CBL-mediated NCE.

## 1. Introduction

The epidermal growth factor (EGF) receptor (EGFR), like other receptor tyrosine kinases (RTKs), regulates key events in cell growth, differentiation, survival and migration [1,2,3]. Aberrant signaling from EGFR has been implicated in many diseases [2,4]. EGFR is historically the prototypical RTK. It was the first of this large family of transmembrane receptors to be cloned, and the first for which a clear connection between aberrant receptor function and cancer could be drawn [2]. The binding of EGF to EGFR at the cell surface induces dimerization of EGFR, which results in the activation of EGFR tyrosine kinase and EGFR trans-autophosphorylation [5,6,7,8,9,10]. EGFR activation stimulates various signaling pathways that regulate multiple cell functions [1,2,3]. EGF stimulates cell proliferation by driving the cell cycle that is comprised of four phases: G1, S, G2 and M [11,12]. Binding of EGF also stimulates the rapid internalization of EGFR [13]. EGFR endocytosis and EGFR-mediated cell signaling are mutually regulated [14,15].

In spite of significant advances in our understanding of EGFR signaling and trafficking, some critical knowledge is still lacking. Our current knowledge of EGFR signaling and EGFR endocytosis comes mostly from the studies of cells in G1 phase of the cell cycle. Very little is known regarding EGFR-mediated signaling and endocytosis in mitosis.

Mitosis represents a period where the needs and requirements of the cell differ vastly from interphase cells. EGFR signaling has been shown to be regulated differently between interphase and mitotic cells. We and others previously found that the EGFR of mitotic cells (at times referred here as “mitotic EGFR” for simplicity) can still be activated during mitosis, but that the signal transduction pathways are regulated differently compared to interphase cells [16,17]. We also previously found that EGFR endocytosis of mitotic cells is regulated differently, in that EGFR is endocytosed at a slower rate [18]. At that time, we did not fully decipher the molecular mechanisms behind the differential kinetics. Therefore, in this report, we further studied this phenomenon.

Endocytosis of the EGFR can lead to two distinct fates for the receptor: recycling back to the plasma membrane or lysosomal degradation [13,19,20,21]. As such, the route taken directly influences the total number of receptors available for a subsequent signal transduction response. EGFR recycling has been shown to be predominantly mediated by clathrin-mediated endocytosis (CME), whereas non-clathrin-mediated endocytosis (NCE) targets receptors for lysosomal degradation [22,23,24].

CME is a mechanism of internalization that is dependent on the recruitment of clathrin to the receptor. While this notion has been disputed by some studies [25,26,27], most data support the theory that CME is inhibited in mitosis [28,29,30,31,32,33,34,35,36,37]. The mechanisms underlying CME inhibition are still unknown, however, several mechanisms have been proposed and partially tested. These mechanisms include the “moonlighting” hypothesis [35,36], the phosphorylation of endocytic proteins [38,39], and the unavailability of actin for CME [40]. In agreement with this, we previously found that mitotic EGFR endocytosis was clathrin-independent as siRNA depletion of clathrin heavy chain did not affect mitotic EGFR endocytosis [18]. We therefore hypothesized that mitotic EGFR proceeded exclusively through NCE.

NCE has been described as having potential tumor suppressive characteristics [41]. NCE has also been described as initiating more slowly than CME [13,22,42,43,44], which fits with our observed delay in mitotic EGFR endocytosis [14]. Molecularly, EGFR NCE has been described as only activated by physiologically high doses of EGF [22,23,24], which is likely a mechanism evolved to compensate when the CME pathway is saturated and to prevent excessive EGFR signaling [38]. EGFR NCE has been shown to be mediated by ubiquitination of the receptor, and this ubiquitination has been shown to be limited by the activity of the E3 ligase c-CBL [22,23,24]. Therefore, c-CBL (henceforth CBL) provides a critical negative regulatory control of the EGFR, as it targets the EGFR for endocytosis and degradation. The activation of CBL depends on its binding to the activated EGFR, either by direct interaction with pY1045, or by indirect interaction through the adaptor GRB2, which binds to pY1068 or pY1086 [23,24,36,37,39,40,45,46].

In this report, we found that EGF-stimulated EGFR endocytosis proceeds exclusively by NCE during mitosis. We found that contrary to interphase cells, mitotic EGFR endocytosis is more reliant on the activation of CBL. Since CME is shut down, CBL has a more crucial role in mediating mitotic EGFR endocytosis. By transfecting HeLa, MCF-7, and 293T cells with CBL siRNA or dominant-negative 70z-CBL, we found that at high EGF doses, CBL is required for EGFR endocytosis of mitotic cells, and not interphase cells. Moreover, the endocytosis of truncated EGFR Δ1044-YFP, which does not bind to CBL, is completely inhibited. EGF induces stronger ubiquitination of mitotic EGFR than interphase EGFR and mitotic EGFR is trafficked to lysosomes for degradation. Furthermore, we found that during mitosis, low doses of EGF also stimulate EGFR endocytosis by NCE. Contrary to interphase, CBL and the CBL-binding regions of EGFR are required for mitotic EGFR endocytosis at low doses. This is due to the mitotic ubiquitination of the EGFR even at low EGF doses.

## 2. Materials and Methods

### 2.1. Antibodies and Chemicals

Antibodies were from Santa Cruz Biotechnology (Santa Cruz, CA, USA), including: mouse anti-EGFR (sc-373746), anti-pY99 (sc-7020), anti-CBL (sc-170), anti-Ubiquitin (sc-8017), anti-Cyclin B1 (sc-245), and anti-β-Tubulin (sc-5274), rabbit anti-GRB2 (sc-8034) and anti-SHC (sc-967), and goat anti pY1068 (sc-16804) and pY1086 (sc-16804). LAMP-2 (AF6228) antibody was from R&D Biosystems (Minneapolis, MN, USA). The horseradish peroxidase (HRP)-conjugated secondary antibodies were from Bio-Rad Laboratories (Hercules, CA, USA) and the fluorescence-conjugated secondary antibodies were from Jackson ImmunoResearch (West Grove, PA, USA). Goat anti-mouse immunoglobulin G (IgG) conjugated with agarose was from Sigma-Aldrich (St. Louis, MO, USA). EGF was from Upstate Biotechnology, Inc. (Lake Placid, NY, USA).

### 2.2. Plasmid Construction

The EGFR-YFP, EGFR-Y1045F-YFP, EGFR-Δ991-YFP, and EGFR-Δ1044-YFP constructs were described previously [47]. The c-CBL-YFP and 70z-CBL-YFP constructs were generous gifts from the Sorkin Lab (Department of Cell Biology, University of Pittsburgh School of Medicine).

### 2.3. Cell Culture, Transfection, and Treatment

HeLa, 293T, and MCF-7 cells were growth at 37 °C in Dulbecco’s modified Eagle’s medium containing 10% fetal bovine serum and antibiotic/antimycotic solution maintained at 5% CO_2_ atmosphere. For transfection, MCF-7 cells in 24-well plates were transfected using LipofectAMINE 2000 reagent (Invitrogen, Carlsbad, CA, USA) as per the manufacturer’s protocol, and 293T cells in 24-well plates were transfected using calcium phosphate precipitation with BES (*N*,*N*-bis(2-hydroxyethyl)-2-aminoethanesulfonic acid) buffer. MCF-7 and 293T cells were chosen due to their low levels of endogenous EGFR [48,49]. Small interfering RNA-mediated silencing transfections were done using CBL siRNA (sc-29241; Santa Cruz Biotechnology, Santa Cruz, CA, USA) in HeLa cells as per the manufacturer’s protocol.

Mitotic cells were collected by gentle mitotic shake-off as previously described [17]. Briefly, cells were arrested in prometaphase by treating cells with nocodazole (200 ng/mL) in serum-free media for 16 h. The nocodazole-arrested cells were treated with EGF (2 ng/mL or 50 ng/mL) for 5, 30, and 45 min, or not treated with EGF (0 min). The EGF-containing media was then removed and serum-free media was added. Cells were placed on ice and dislodged by gently tapping the plates for 5 min. The mitotic cell-containing media was centrifuged at 1000 rpm for 5 min. The obtained mitotic cells were then lysed with cold Mammalian Protein Extraction Reagent (M-Per) (Thermo Fisher Scientific Inc, Rockford, IL, USA) buffer in the presence of phosphatase and protease inhibitors including 100 mm NaF, 5 mM MgCl_2_, 0.5 mM Na_3_VO_4_, 0.02% NaN_3_, 0.1 mM 4-(2-aminoethyl)-benzenesulfonyl fluoride, 10 μg/mL aprotinin, and 1 μM pepstatin A. To collect lysates for interphase cells, cells were serum-starved for 16 h. Cells were then treated with EGFR for 5, 30, and 45 min. To ensure consistency with the mitotic treatment, the cells were also tapped on ice for 5 min to remove mitotic cells and then left on ice for 5 min. The remaining interphase cells were collected by scraping on ice in cold M-Per in the presence of phosphatase and protease inhibitors. For both interphase and mitotic cells, after lysing, the samples were centrifuged at 21,000× *g* and the supernatant was collected for immunoblotting.

### 2.4. Immunoprecipitation and Immunoblotting

Immunoprecipitation (IP) experiments were carried out as described previously [47] Interphase or mitotic cells were lysed with IP buffer (20 mM Tris, pH 7.5, 150 mM NaCl, 1% NP40, 0.1% sodium deoxycholate, 100 mM NaF, 0.5 mM Na_3_VO_4_, 0.02% NaN3, 0.1 mM 4-(2-aminoethyl)-benzenesulfonyl fluoride, 10 μg/mL aprotinin, and 1 μM pepstatin A) for 15 min at 4 °C. Cell lysates were then centrifuged at 21,000× *g*. The supernatant, containing 1 mg of total protein, was incubated with 0.8 μg of mouse monoclonal anti-EGFR antibody A-10 (Santa Cruz Biotechnology, Santa Cruz, CA, USA) for 2 h at 4 °C with gentle mixing by inversion. Goat anti-mouse IgG conjugated with agarose was added to each fraction and incubated for 2 h at 4 °C with gentle mixing by inversion. Next, the agarose beads were centrifuged, washed three times with IP buffer, and 2× loading buffer was added. The samples were boiled for 5 min at 95 °C and loaded for SDS-PAGE for subsequent immunoblotting.

Immunoblotting was performed as previously described [50]. Briefly, protein samples were separated by SDS-PAGE and were transferred to nitrocellulose. The membranes were blocked for non-specific binding and incubated with primary antibody overnight. The membranes were then probed with HRP-conjugated secondary antibody followed by detection with enhanced chemiluminescence solution (Pierce Chemical, Rockford, IL, USA) and light detection on Fuji Super RX Film (Tokyo, Japan).

### 2.5. Indirect Immunofluorescence

Indirect immunofluorescence was performed as previously described [18]. Cells were grown on glass coverslips and serum-starved for 16 h. After treatment without or with nocodazole (200 ng/mL for 16 h) and without or with EGF for various indicated times, the cells were fixed with ice-cold methanol for 10 min. The cells were then permeabilized with 0.2% Triton X-100 for 10 min on ice. Next, cells were blocked with 1% BSA for 1 h on ice. Cells were then incubated with primary antibody overnight at 4 °C. Primary antibody anti-CBL was used at 1:50 dilution, and anti-pEGFR-Y1086 and anti-EGFR were used at 1:200 dilutions. Cells were then washed three times with PBS and incubated with rhodamine- or FITC-labeled secondary antibody for 1 h at 4 °C. Cells were then washed three times with PBS, followed by nuclear staining with DAPI (4′6-diamidino-2-phenylindole) (300 nM). Finally, cells were washed three times and mounted. Images were taken with the DeltaVision deconvolution microscopy system (GE Healthcare Life Sciences, Buckinghamshire, UK) or Axiovert 200 inverted microscopy system (Carl Zeiss, Oberkochen, Germany)

Quantification of EGFR internalization was performed using ImageJ as previously described [18]. Briefly, the cells were visualized by differential interference contrast (DIC). For each image, a large polygon (VL) was drawn along the outer edge of the cell membrane to represent the entire area of the cell. In addition, a small polygon (VS) was drawn along the inner edge of the cell membrane to represent the cell interior. The VL and VS values were calculated for either stains of EGFR, pEGFR, or for YFP (for EGFR-YFP mutants), and membrane EGFR percentage was obtained by the Equation (1):(1)$$\mathrm{Membrane}\;\mathrm{EGFR}\;\mathrm{percentage}=\;\frac{V_{L}-V_{S}}{V_{L}}$$

The membrane EGFR percentage was calculated and plotted for at least 10 cells in at least two independent experiments. Standard errors of the mean (SEM) are displayed for each data point. Statistics were performed by two-tailed Student’s t-test (** indicates *p* < 0.01 and * indicates *p* < 0.05).

## 3. Results

### 3.1. CBL Interaction with EGFR during Mitosis

EGFR expression at the plasma membrane does not change from interphase to mitosis [17,18,51]. Previously, we found that similar to interphase, stimulation of nocodazole-arrested mitotic HeLa cells with high doses of EGF (50 ng/mL) induces the phosphorylation of the EGFR at all major tyrosine residues, including Y992, Y1045, Y1068, Y1086, and Y1173 [17]. Moreover, this also phosphorylates CBL to similar levels [17].

It has been well shown that EGF stimulates CBL E3-ligase activity [52,53]. Phosphorylated EGFR creates docking sites for CBL to translocate from the cytoplasm to the plasma membrane and ubiquitinate EGFR. Therefore, to confirm mitotic CBL activation by EGF stimulation, we first observed CBL localization in mitotic HeLa cells by immunofluorescence microscopy. Immunofluorescence costaining using anti-EGFR and anti-CBL antibodies revealed that CBL colocalizes with EGFR upon 5 min of 50 ng/mL EGF treatment in both interphase and mitotic cells (Figure 1A). Furthermore, IP of EGFR using a monoclonal anti-EGFR antibody of both interphase and mitotic cell lysates showed that mitotic cells stimulated with EGF for 5 min not only co-immunoprecipitated CBL, but also had higher IPs of CBL with EGFR than interphase cells (Figure 1B). Interestingly, CBL co-immunoprecipitation (co-IP) with EGFR decreased at 30 min after EGF treatment in mitotic cells, whereas it increased for interphase cells, and continued increasing at 45 min after EGF treatment. Most surprising, however, is that ubiquitination of the EGFR was enhanced at all time points studied during mitosis compared to interphase (Figure 1B). Since CBL also binds EGFR indirectly through the EGFR adaptor GRB2, we also immunoblotted EGFR co-immunoprecipitates for GRB2 and SHC. The results showed that during mitosis, GRB2 and SHC also bind to EGFR following EGF stimulation (Figure 1B).

In summary, double indirect immunofluorescence revealed that both EGFR and CBL co-localize after EGF stimulation during mitosis. Co-IP experiments also showed that EGF stimulates the interaction between EGFR and CBL. In addition, EGFR is more strongly ubiquitinated by EGF stimulation during mitosis.

### 3.2. Effects of Altering CBL Expression and Activity during Mitosis

CME has been shown to be inhibited during mitosis [37,39,40]. Therefore, we sought to discover whether altering CBL activity, the major mediator of NCE, would inhibit EGFR endocytosis. We first silenced CBL in HeLa cells by siRNA transfection and found that transfected mitotic cells had much less EGFR endocytosis following 45 min of EGF (50 ng/mL) stimulation, as observed by immunofluorescence (IF) staining of activated EGFR (Figure 2A). In comparison, siRNA-transfected interphase cells were little affected. To further confirm the effects of CBL knockdown without the use of immunofluorescence staining, MCF-7 cells were transiently transfected with EGFR-YFP, and YFP localization was monitored. MCF-7 cells were selected due to their low level of endogenous EGFR expression [54]. Similarly, knockdown of CBL by siRNA in MCF-7 cells also inhibited mitotic endocytosis of EGFR-YFP following 45 min of EGF (50 ng/mL) stimulation (Figure 2B).

To further verify the role of CBL, we used the dominant-negative 70Z-CBL-YFP mutant, which has a deletion of 17 amino acids that disrupts the RING finger structure, making it unable to interact properly with ubiquitin-conjugating enzymes (E2 ligases) [53,55]. The 70Z-CBL-YFP protein can still bind to the cytoplasmic tail of activated EGFR [55,56,57,58]. Transfection with 70Z-CBL-YFP showed that 70Z-CBL-YFP became localized to EGFR following EGF stimulation (50 ng/mL), and that this significantly inhibited EGF-induced EGFR endocytosis during mitosis, but not in interphase, where 70Z-CBL-YFP transfection did not change EGF-induced endocytosis (Figure 3A). Quantification of the data from Figure 3A showed that mitotic cells transfected with 70Z-CBL-YFP retained EGFR at the plasma membrane when compared with non-transfected cells, even after 60 min of EGF treatment (50 ng/mL) (Figure 3B). Taken together, inhibiting CBL activity decreased EGFR endocytosis in mitosis, but not in interphase. Therefore, CBL activity appears more important during mitotic EGFR endocytosis than during interphase.

We next sought to see whether CBL overexpression could increase the rate of endocytosis during mitosis. CBL-YFP overexpression in HeLa cells did not induce any observable endocytosis at earlier time points, nor increase the rate of EGFR internalization, when we observed EGFR endocytosis of these cells at 2, 5, 7, 10, or 15 min (Figure 4). EGFR endocytosis of CBL-YFP-transfected cells at 30, 45, and 60 min also progressed similarly to non-transfected cells (Appendix A). This is similar to previous reports in interphase cells, where it was observed that overexpression of CBL did not increase the rate of EGFR internalization [57,59].

### 3.3. Role of EGFR C-Terminal Domains for Mitotic Endocytosis

We previously showed that mitotic EGFR endocytosis requires EGFR kinase activity [18]. Treatment with the EGFR-tyrosine kinase antagonist AG1478 inhibited mitotic EGFR endocytosis, and washed away AG1478-restored endocytosis [18]. In contrast, interphase EGFR could still undergo endocytosis in the presence of AG1478 [18]. To further explore the role of EGFR kinase activity in the activation of CBL, we blocked EGFR activation with AG1478 for 1 h prior to EGF (50 ng/mL) treatment. As before, this treatment prevented EGFR endocytosis during mitosis as visualized by IF [18] (data not shown). We next examined CBL phosphorylation by immunoblotting with antibody to p-CBL and showed that treatment with AG1478 inhibited EGF-induced CBL tyrosine phosphorylation in mitotic cells (Figure 5A). Therefore, EGFR kinase activity is required for CBL activation.

We next sought to investigate which EGFR domains were important for mitotic endocytosis. We made use of previously constructed and characterized YFP-tagged EGFR mutants including EGFR with Y1045F substitution (Y1045F-YFP, no direct CBL binding), EGFR truncated at 1045 (Δ1044-YFP, no CBL binding), EGFR truncated at 992 (Δ991-YFP, no internalization), and WT (EGFR-YFP) (Figure 6F) [47,55,60,61,62]. We transfected these constructs into HEK 293T or MCF-7 cells, since they express low amounts of endogenous EGFR, and then observed the effects of EGF treatment on their plasma membrane localization using indirect immunofluorescence (Figure 5B–E and Figure 6A–E). 

In non-EGF-treated MCF-7 cells, all mutants exhibited high plasma membrane localization and low cytoplasmic localization during both interphase and mitosis (Figure 6A–E). For the cells in the interphase, treatments of EGF (50 ng/mL) for 30, 45, or 60 min significantly increased the internalization of EGFR-YFP, Y1045F-YFP, and Δ1044-YFP, but not Δ991-YFP which is endocytosis deficient due to the lack of any internalization motifs. The internalization levels of EGFR-YFP, Y1045F-YFP, and Δ1044-YFP during interphase at all three time points were all similar. In contrast, these mutants responded to EGF treatment differently from each other when cells were in mitosis. For the cells in mitosis, EGF stimulated strong endocytosis of EGFR-YFP, and approximately two-thirds of EGFR-YFP was internalized following 30 min of EGF treatment, with more EGFR-YFP becoming internalized at 45 and 60 min. The internalization of both Y1045F-YFP and Δ1044-YFP, however, were impaired in mitosis. In mitosis, no endocytosis of Y1045F-YFP mutants was observed following addition of EGF for 30 min, with only a very low level of endocytosis at 45 min. Interestingly, a high proportion of them eventually became endocytosed at 60 min. However, no EGF-induced endocytosis of Δ1044-YFP mutants was observed even at 60 min following EGF addition (Figure 6). Similar results were observed when the experiments were repeated in 293T cells (Figure 5B–E).

Taken together, this data shows that the CBL-binding domains of the EGFR are more important for mitotic EGFR endocytosis than interphase. These results also suggested that GRB2 cooperation for indirect CBL-binding to EGFR dramatically increases mitotic EGFR endocytosis.

### 3.4. Endocytic Trafficking of EGFR in Mitosis

The endocytic pathway that the EGFR takes has been shown to influence the fate of the EGFR. CME has been shown to lead predominantly to EGFR recycling, whereas NCE targets receptors for lysosomal degradation [23,24]. Since we observed that EGFR endocytosis during mitosis proceeds exclusively in CBL-mediated NCE, we hypothesized that EGFR endocytosis during mitosis should lead exclusively to lysosomal trafficking. To test this, we examined the colocalization of endocytic route markers with EGFR by fluorescence microscopy. The EGFR of both mitotic and interphase cells showed strong colocalization with EEA-1 and RAB5 after 30 min of EGF treatment, indicating that the EGFR is trafficked to early endosomes (Figure 7). EEA-1 and RAB5 did not colocalize with any EGFR at the plasma membrane of either mitotic or interphase cells. Staining with the late endosomal markers LAMP-2 showed that EGFR and LAMP-2 colocalized after EGF stimulation for 60 min. These data indicated that EGFR was targeted to lysosomes through NCE during mitosis.

### 3.5. Low EGF Doses Activate Mitotic EGFR NCE

The above experiments were all performed using high concentrations of EGF (50 ng/mL). During interphase, low doses of EGF only activates CME, whereas high doses activate both CME and NCE [22,23,24]. If this finding is also applied to mitosis, low doses of EGF should not induce EGFR endocytosis in mitosis as CME is inhibited in mitosis and only high doses of EGF activate NCE. However, we previously observed that low doses of Texas Red-conjugated EGF (TR-EGF) (2 ng/mL) could still lead to their internalization in mitotic HeLa and CHO cells in a similar pattern as high doses of EGF [18,14]. We therefore decided to address this discrepancy.

We hypothesized that similar to high-dose EGF, low-dose EGF is still able to stimulate EGFR endocytose through NCE in mitosis. To test this hypothesis, we first repeated experiments described in Figure 3, Figure 4, Figure 5 and Figure 6 with a low dose of EGF (2 ng/mL), and we indeed obtained similar results (Figure 8 and Figure 9). As shown in Figure 8, in MCF-7 cells transfected with EGFR-YFP, EGF at the low dose (2 ng/mL) induced EGFR endocytosis in both interphase and mitotic cells (Figure 8). The endocytosis of EGFR-Y1045F-YFP was only observed at 60 min following EGF addition in mitosis, and EGFR-Δ1044-YFP was deficient in endocytosis during mitosis (Figure 8). These data suggest that at low-dose EGF, mitotic EGFR endocytosis requires its interaction with CBL. To further determine the involvement of CBL in low-dose EGF-stimulated EGFR endocytosis in mitosis, we transfected Hela cells with non-functional 70z-CBL-YFP (Figure 9). As in high-dose EGF conditions, expression of 70z-CBL-YFP inhibited EGFR endocytosis in response to low-dose EGF (Figure 9A–B). Together, these data suggest that EGFR endocytosis induced by low-dose EGF is also mediated by CBL. On the other hand, overexpression of CBL-YFP again did not affect EGFR endocytosis in mitosis in response to low-dose EGF (Figure 9C), with endocytosis beginning around 15 min for transfected and non-transfected cells.

We next sought to uncover why low-dose EGF is able to stimulate EGFR endocytosis through CBL-mediated NCE in mitosis, but not in interphase. We therefore examined whether this is due to CBL-mediated ubiquitination of EGFR. To this end, we treated Hela cells with EGF at 2 ng/mL or 50 ng/mL for 45 min and examined the ubiquitination of EGFR in both interphase and mitosis (Figure 10A). Co-IP of EGFR and immunoblotting for ubiquitin revealed that as before, high-dose EGF (50 ng/mL) induced higher EGFR ubiquitination in mitosis more than in interphase. Low-dose EGF (2 ng/mL) did not induce the ubiquitination of EGFR in interphase, as previously reported [22,23,24]. However, low-dose EGF stimulation caused significant EGFR ubiquitination in mitotic cells. Moreover, the binding of CBL to EGFR followed the same pattern as ubiquitination, with CBL again binding to EGFR at low doses during mitosis, but not during interphase.

We also performed similar experiments with different times of EGF stimulation, at 5, 30, and 45 min (Figure 10B). The phosphotyrosine-specific antibody pY99 was used to confirm EGFR phosphorylation. Blotting for ubiquitin revealed that during mitosis ubiquitination of EGFR occurred at 5 min and was sustained through to 45 mins in response to low-dose EGF stimulation. In contrast, in interphase, EGFR ubiquitination only occurred briefly at 5 min, and ubiquitination was very weak at later time points. In addition, CBL and SHC were pulled-down with EGFR during both interphase and mitotic. Therefore, it appears that low-dose EGF stimulation has different effects on EGFR ubiquitinates in mitosis from interphase.

Since mitotic EGFR is strongly ubiquitinated at low doses of EGF, and ubiquitination has been associated with EGFR degradation, we hypothesized that low EGF doses could lead to EGFR degradation during mitosis. Total cell lysates of interphase and mitotic cells treated with low doses of and total EGFR levels were assayed by the western blot. Whereas interphase EGFR levels remain constant throughout 45 min of low-dose EGF treatment, we found that mitotic EGFR levels drop drastically with time (Figure 10C). Taken together, these results suggested that, unlike interphase, low doses of EGF activate CBL-mediated EGFR degradation in mitotic cells. Interestingly, by the western blot, the CBL band appears smaller in mitotic samples than in interphase samples.

## 4. Discussion

Our results showed that EGF-induced EGFR endocytosis during mitosis proceeds exclusively by CBL-dependent NCE (Figure 11). NCE plays a major role in the regulation of EGFR fate by targeting it to lysosomes for degradation. Our research has uncovered a temporal period during which the cell likely exclusively targets EGFR for degradation. This bypasses the receptor recycling pathway that is undesirable if the goal is EGFR attenuation, or if it is to deliver and keep a pharmacological agent into a cell [63]. Targeting mitotic cells is feasible for EGFR-overexpressing cancer cells, as these cells intrinsically undergo more cell proliferation. In addition, the population of mitotic cells can be increased by treatment with anti-mitotic drugs, such as the commonly used taxanes and vinca alkaloids. Therefore, mitotic cells of EGFR-overexpressing cells can be targeted more directly. Moreover, the FDA-approved EGFR antibody cetuximab has been shown to initiate receptor endocytosis [64]. Whether mitotic EGFR treated with EGFR antibodies is also internalized by NCE remains to be investigated. However, if it does, nano-conjugation of EGFR antibodies to pharmacological agents may provide a targeted approach to treating these cancers.

The study of EGFR NCE, thus far, has relied on the inhibition of clathrin, as well as the use of high doses of EGF to activate NCE. Our results indicated that mitotic EGFR endocytosis is exclusive through NCE. Thus, mitotic cells offer an alternate system for studying the NCE of the EGFR, regardless of EGF dosage. In general, NCE is much more complicated and very little understood. It is no surprising that very little is known regarding the regulation of mitotic NCE of EGFR. Our findings that mitotic EGFR endocytosis is mediated by CBL through NCE at both high and low doses of EGF advances our understanding on both EGFR endocytosis and NCE in general.

Low doses of EGF have been shown to only activate CME, while also causing CBL to bind EGFR. The CBL bound to the EGFR is also able to ubiquitinate EGFR. How is it that this does not induce NCE? The answer is that with low EGF doses, the EGFR is not ubiquitinated sufficiently to undergo NCE. A rather exquisite model to explain how CME and NCE are regulated by EGF dosage threshold has been put forward [23]. According to the experimentally validated model of Sigismund and colleagues, low doses of EGF around 1 ng/mL scarcely phosphorylate the EGFR, causing a low probability that CBL alone or CBL/GRB2 complex can bind the receptor, and as such translating into low EGFR ubiquitination. Low EGFR ubiquitination does not activate NCE, and endocytosis proceeds by CME. However, as EGF doses increase, the phosphorylation of each of the three EGFR tyrosine residues increases gradually. Then, for example, if the EGF dosage only catalyzes the phosphorylation of pY1045, and not pY1068 or pY1086, a rather unstable CBL binding occurs to the EGFR, leading to some but low ubiquitination of the EGFR. However, if all three tyrosine sites are phosphorylated, a highly stable GRB2/CBL binding can form on the EGFR, thus strongly ubiquitinating the EGFR. The kinetics involved in the probability of the CBL/GRB2 complex binding to each combination of pY site predicts a sharp rise over a narrow range of EGF dose, effectively causing EGFR ubiquitination to rise sharply once a certain dosage of EGF is applied.

The theory that EGFR ubiquitination is absolutely necessary for endocytosis has been a subject of controversy [56,61,65,66]. Our research supports the notion that ubiquitination by CBL is important for NCE [22,23]. Furthermore, our research provides strong support for the requirement of CBL and GRB2 binding to the EGFR in order to cause its ubiquitination [23,24]. Our results argue that GRB2-mediated CBL binding is more important than direct CBL-binding during mitosis, as the internalization of the Δ1044 mutant is significantly inhibited, whereas the internalization of the Y1045F mutant is only slightly inhibited. Overexpression of CBL did not accelerate nor enhance mitotic EGFR endocytosis in response to EGF. This has also been recently reported in vivo [67]. NCE has been reported to initiate more slowly than CME [13,22,42,44,55,68], and it therefore appears that CBL overexpression is not the limiting factor in the speed of EGFR NCE. Other important mediators of NCE, such as EPS15, EPS15R, and EPSIN [22], or endoplasmic reticulum (ER)-resident protein reticulon 3 (RTN3) and CD147 [41], may warrant investigation. More importantly, the exact mechanism by which the ubiquitination of the EGFR induces internalization is still unknown, and studies to elucidate the precise molecular mechanism would be extremely impactful.

The inactivation of CBL has been shown to display pro-oncogenic features [69,70,71,72,73]. Moreover, common pro-oncogenic EGFR mutations L858R and L858R/T790M have impaired CBL-binding, slower endocytosis, and impaired degradation [74]. EGFRvIII, the most common variant in gliomas, also has a reduced interaction with CBL and thus impaired ubiquitination owing to hypophosphorylation of pY1045 [75,76]. Here, we showed that CBL activity during mitosis is even more important, and its activity is enhanced compared to interphase cells. Evolutionarily, since mitotic cells do not have active CME [37,39], the activation of NCE may have been even more critical during mitosis to suppress EGFR overactivation. A loss of CBL activity, whether by inactivating mutations to CBL or EGFR CBL-binding, would therefore have a more pronounced effect during mitosis, as the EGFR would continue to signal excessively. The functional role of mitotic EGFR activation is still not well known. It is unclear how abnormally sustained EGFR signaling during mitosis affects cellular processes, however, it does appear to help mitotic cancer cells resist nocodazole-mediated cell death [17].

We showed that the EGFR is more strongly ubiquitinated during mitosis at both low and high doses of EGF, suggesting that CBL activity is enhanced during mitosis. How can CBL be better primed to induce endocytosis, even at low concentrations of EGF during mitosis? It has been shown that CBL also acts as an adaptor in the CME pathway. Since CME is no longer active during mitosis, a possible explanation may that due to increased CBL protein availability, the cellular pool of CBL no longer needs to divide its time between CME and NCE. Another explanation may be that CBL is modified during mitosis to be better primed for its E3 ligase activity. Interestingly, probing with the CBL antibody revealed that the mitotic CBL band appears less prominent compared to interphase (Figure 9), and IF CBL staining appears weaker in mitotic cells as well (Figure 1A), although the exact significance is unknown. Another alternative possibility may revolve around the DUBs (deubiquitinating enzymes) that deubiquitinate EGFR. Fifteen DUBs have been reported to impact EGFR fate, although some may be deubiquitinating non-EGFR component, such as EPS15 [77]. These DUBs could be shut off during mitosis, causing ubiquitination to persist longer than during the interphase.

Mitosis represents a phase of tremendous transition to the cell. It is a critical moment of the cell’s life where its genetic material is precisely separated into two daughter cells. To ensure proper chromosome segregation, the mitotic environment must be carefully controlled, as improper mitosis results in chromosome bridging, lagging, or aggregation, and ultimately aneuploidy [78]. To achieve this, mitotic cells undergo mitotic cell rounding, which is the dramatic transformation of cells from well spread and flattened to spherical and rigid. The mitotic shape is thought to confer cells with a predictable and defined geometry, regardless of its external environment, so as to facilitate chromosome capture and the symmetric segregation of contents [79,80,81]. The spherical shape and rigid cell cortex, however, present the EGFR with changes in conditions during mitosis. For example, it is likely that the inhibition of CME during mitosis is due to the unavailability of actin in participating in endocytosis, as it must form the rigid mitotic cell cortex [40]. Furthermore, mitosis changes the amount of space in the cell, so that they reach a minimal volume during metaphase [82]. It is possible that the molecular interactions necessary for NCE may become facilitated by a smaller cell volume. The more compacted volume does not appear to aid in EGFR activation, as the phosphorylation of the EGFR upon EGF addition is similar between interphase cells and mitotic cells. Therefore, in this situation, the smaller cell volume may aid CBL or GRB2 binding to the EGFR.

Another change to normal interphase EGFR signaling during mitosis is the global phosphorylation of mitotic proteins by mitotic kinases. For example, studies have shown the mitotic phosphorylation of over 1000–6027 proteins, including 14,000–50,000 phosphorylation events depending on the study [83,84,85]. Interestingly, many phospho-sites overlap between EGF-stimulated cells and mitotic cells [85]. Indeed, various components of the EGFR signaling and endocytic pathways appear to play different roles in mitosis, a phenomenon known as moonlighting [38,86]. This includes important members of EGFR CME, such as clathrin, dynamin, and AP-2 [38,87,88,89,90]. Since these proteins and many others are moonlighting in mitosis-related processes, their availability to participate in EGFR endocytosis during mitosis may be compromised. This may also affect EGFR signaling. It has been shown that EGF-induced AKT activation requires EGFR residence in clathrin-coated pits, but not internalization [91,92]. We previously showed that only AKT2, and not AKT1, becomes activated following EGF stimulation during mitosis [17]. Since CME is shut down during mitosis, it can be speculated that the differential activation of AKT during mitosis is a consequence of the inability of clathrin to be involved in mitotic EGFR endocytosis. Therefore, the changes imparted by global mitotic phosphorylation and mitotic cell rounding cannot be discounted to EGFR signaling, and likely of other signaling receptors as well.

In our study, we made use of the microtubule depolymerizer nocodazole to arrest cells in mitosis. So far, nocodazole has still been the most widely used drug for arresting cells in mitosis [85,93,94,95]. We decided to use nocodazole in our research in order to obtain synchrony between our western blots, co-IPs, and immunofluorescence experiments, as it has been shown that the sub-stage of mitosis can influence the kinetics of endocytosis [18]. Previous research has shown that factors, such as serum starvation, nocodazole, and other mitotic inhibitors, could inhibit CME [96]. However, it should be noted that the researchers were evaluating transferrin receptors, which is endocytosed by constitutive endocytosis rather than the ligand-induced mechanism used by EGFR. Furthermore, our previous study that showed that clathrin downregulation by siRNA has no effect on mitotic EGFR endocytosis performed without the use of nocodazole [18]. We have also previously shown that 16 h nocodazole treatment does not lead to significant cell apoptosis [17].

However, nocodazole is a microtubule depolymerizer, and it is possible that it may interfere with components of the endocytic pathway. It has been reported that nocodazole blocks the transport from early to late endosomes, as this trafficking may involve microtubules. Transport of material from early to late endosomes was shown to be inhibited by nocodazole for fluid-phase endocytosis of dextran and constitutive endocytosis of transferrin in HeLa cells [97,98]. Moreover, in MEK cells, nocodazole reduces the percentage of EGFR colocalizing to LAMP-2-containing vesicles [99]. It may therefore be possible that the stronger ubiquitination seen on the EGFR is due to some inability of ubiquitinated EGFR to be trafficked to lysosomes for degradation. However, our results showed that the EGFR of nocodazole-arrested mitotic cells can colocalize with LAMP-2 after 60 min of high-dose EGF treatment (Figure 7C). The microtubule stabilizer paclitaxel promotes EGFR degradation to lysosomes in A549 cells, apparently due to a spatially shorter route for the EGFR to lysosomes [100]. Additionally, nocodazole treatment does not change the distribution of lysosomal membranes [101]. Regardless, our results may be more representative of drug-arrested mitosis.

The EGFR uses various signaling pathways to achieve numerous pro-oncogenic cellular outcomes. Endocytosis downregulates EGFR signaling from the cell surface, but initiates intracellular signaling from endosomes [102]. In this way, endocytosis controls EGFR signaling, spatially and temporally, making it an indispensable part of receptor signaling. Therefore, the interplay between EGFR signaling and endocytosis critically determines cellular outcome.

## 5. Conclusions

In conclusion, our research further showed that mitotic EGFR endocytosis proceeds through CBL-mediated NCE, which supports the notion that mitotic endocytosis is not completely inhibited, but proceeds through NCE. The unique property of mitotic EGFR endocytosis offers an important opportunity for developing cancer therapy that targets both EGFR and mitosis.

## Figures and Tables

**Figure 1 cells-07-00257-f001:**
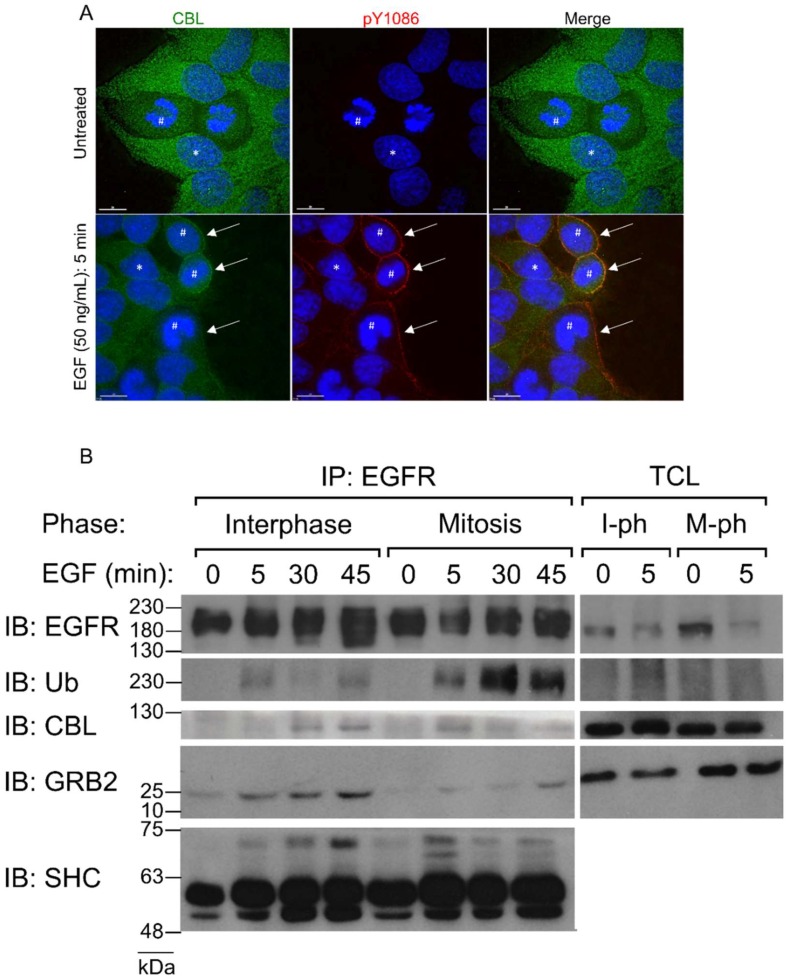
CBL is activated by EGF stimulation during mitosis. (**A**) Direct immunofluorescence images of HeLa cells stained with CBL (green), EGFR pY1086 (red), and DAPI (blue). Cells were untreated or treated with EGF (50 ng/mL) for 5 min. * represents interphase cells and # represents mitotic cells. Arrows point to sites of CBL colocalization to EGFR in mitotic cells. (**B**) Co-immunoprecipitation of EGFR from asynchronous (interphase) or nocodazole-arrested (mitosis) HeLa cells. EGF (50 ng/mL) was used to treat cells for the indicated times. Immunoblotting was performed with the specified antibodies. Mitotic EGFR is more strongly ubiquitinated than interphase. Total cell lysates (TCLs, or input) are also shown. Results are representative of at least two biological replicates. IB: Immunoblot. Ub: Ubiquitin. GRB2: Growth factor receptor-bound protein 2. SHC: Src homology 2 domain containing. IP: Immunoprecipitate. I-ph: Interphase/M-ph: Mitosis phase.

**Figure 2 cells-07-00257-f002:**
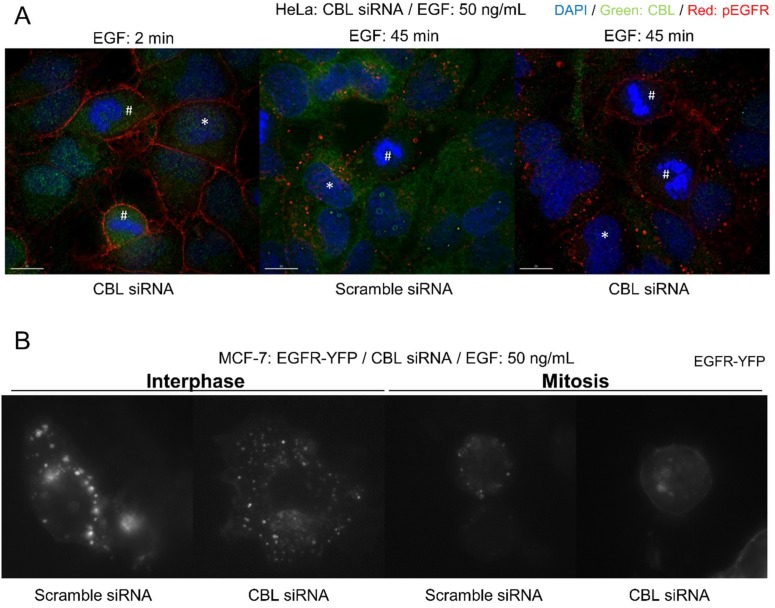
siRNA downregulation CBL inhibits mitotic endocytosis. Indirect immunofluorescence to observe EGFR endocytosis in cells treated with CBL siRNA or with scramble siRNA in: (**A**) HeLa cells stained for DAPI (blue), CBL (green) and phosphorylated EGFR (pY1086) (red) and treated with EGF (50 ng/mL) for 45 min; and (**B**) MCF-7 cells transfected with EGFR-YFP and treated with EGF (50 ng/mL) for 45 min. MCF-7 cells were treated with nocodazole (200 ng/mL) for 16 h. * represents interphase cells and # represents mitotic cells.

**Figure 3 cells-07-00257-f003:**
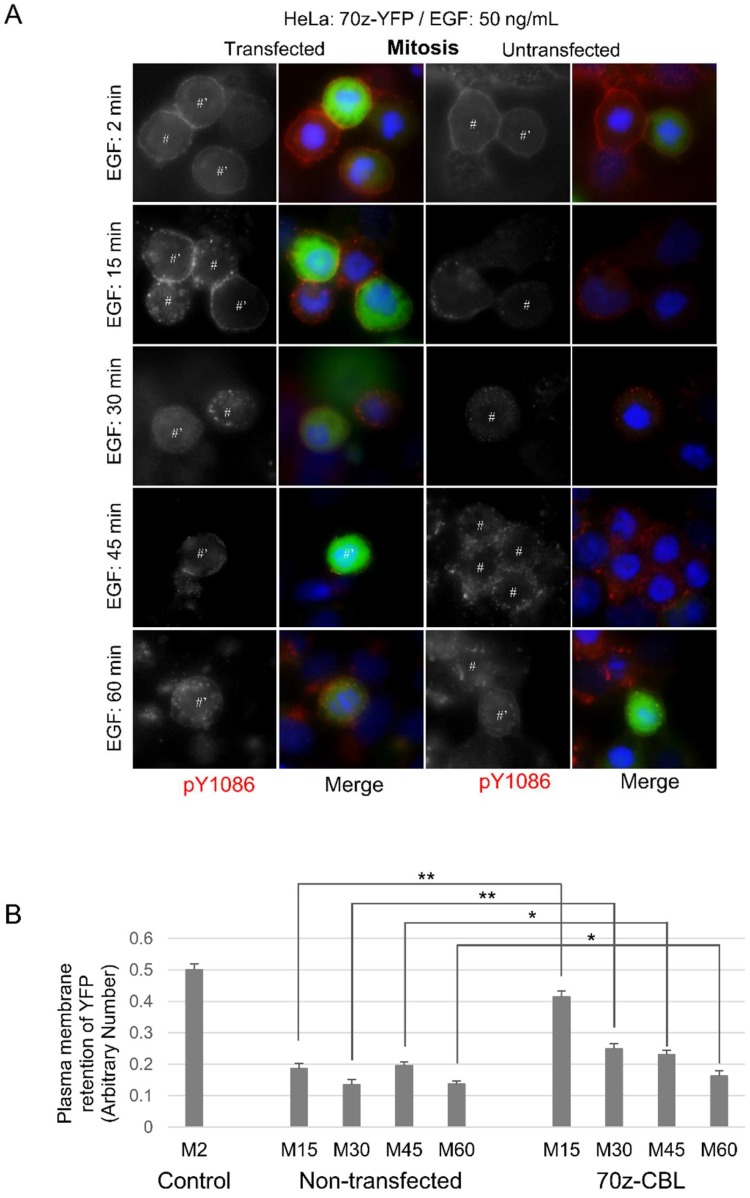
Dominant-negative CBL inhibits mitotic endocytosis. Indirect immunofluorescence to observe EGFR endocytosis in HeLa cells transfected with dominant-negative CBL (70z-YFP). Following the transfection of CBL, the cells were treated with nocodazole (200 ng/mL) for 16 h. The cells were then treated with EGF (50 ng/mL) for the indicated times and were stained for pY1086 (red), and DAPI (blue). The transfected cells were green. (**A**) EGFR endocytosis in cells transfected with 70z-YFP. * represents interphase cells, # represents mitotic cells, and ’ represents transfected cells. (**B**) Quantification of EGFR retained in the plasma membrane in mitotic cells from experiments described in (**A**). Each data is the average of at least 10 mitotic cells. Control are mitotic cells treated with EGF (50 ng/mL) for 2 min and with fully plasma membrane-localized EGFR. The error bars are the standard errors of the mean (SEMs).

**Figure 4 cells-07-00257-f004:**
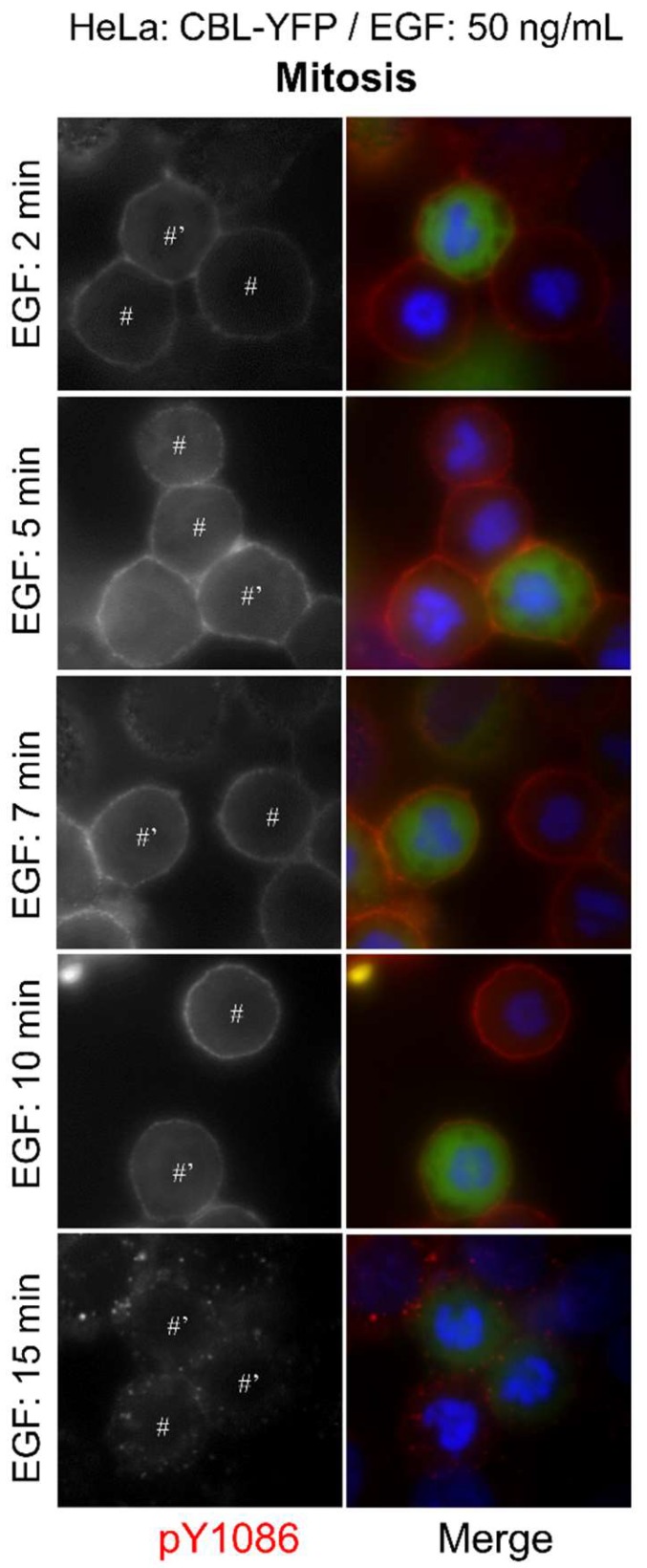
Overexpression of CBL does not accelerate endocytosis. Indirect immunofluorescence to observe EGFR endocytosis in HeLa cells transfected with CBL (CBL-YFP). Following the transfection of CBL, the cells were treated with nocodazole (200 ng/mL) for 16 h. The cells were then treated with EGF (50 ng/mL) for the indicated times and were stained for pY1086 (red), and DAPI (blue). The transfected cells were green. The endocytosis of EGFR in the cells transfected with wild type c-CBL-YFP. * represents interphase cells, # represents mitotic cells, and ’ represents transfected cells.

**Figure 5 cells-07-00257-f005:**
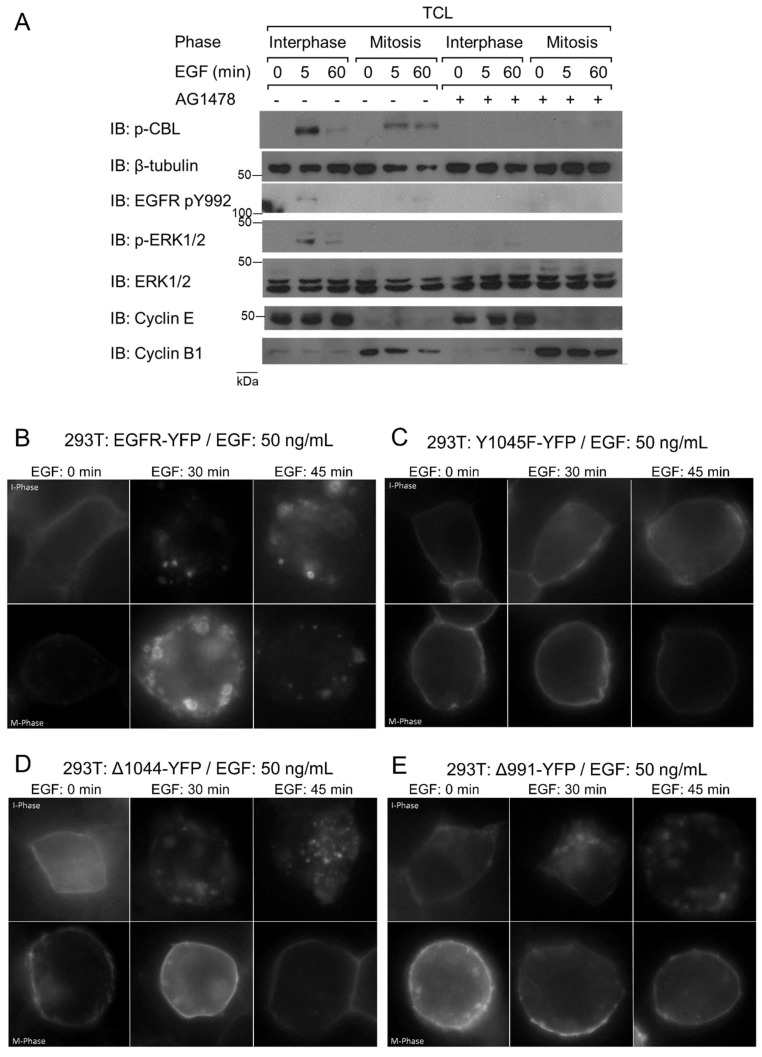
The role of EGFR kinase activation in EGFR endocytosis. (**A**) The effects of AG1478 on EGFR and CBL phosphorylation. Asynchronous (I-phase) and nocodazole-arrested (M-phase) HeLa cells were pre-treated with AG1478 1 h prior to EGF treatment, and then treated with EGF (50 ng/mL) for the indicated times. Western blots of TCL are shown to study the role of EGFR C-terminal domains, and 293T cells were transfected with (**B**) EGFR-YFP (positive control), (**C**) EGFR-Y1045F-YFP (no direct CBL binding), (**D**) EGFR-Δ1044-YFP (no CBL binding), and (**E**) EGFR-Δ991-YFP (negative control). Cells were treated with EGF (50 ng/mL) for the specified times and observed by indirect immunofluorescence. Cell cycle phase of cells were determined by DNA morphology (not shown). Cells were treated with nocodazole (200 ng/mL) for 16 h.

**Figure 6 cells-07-00257-f006:**
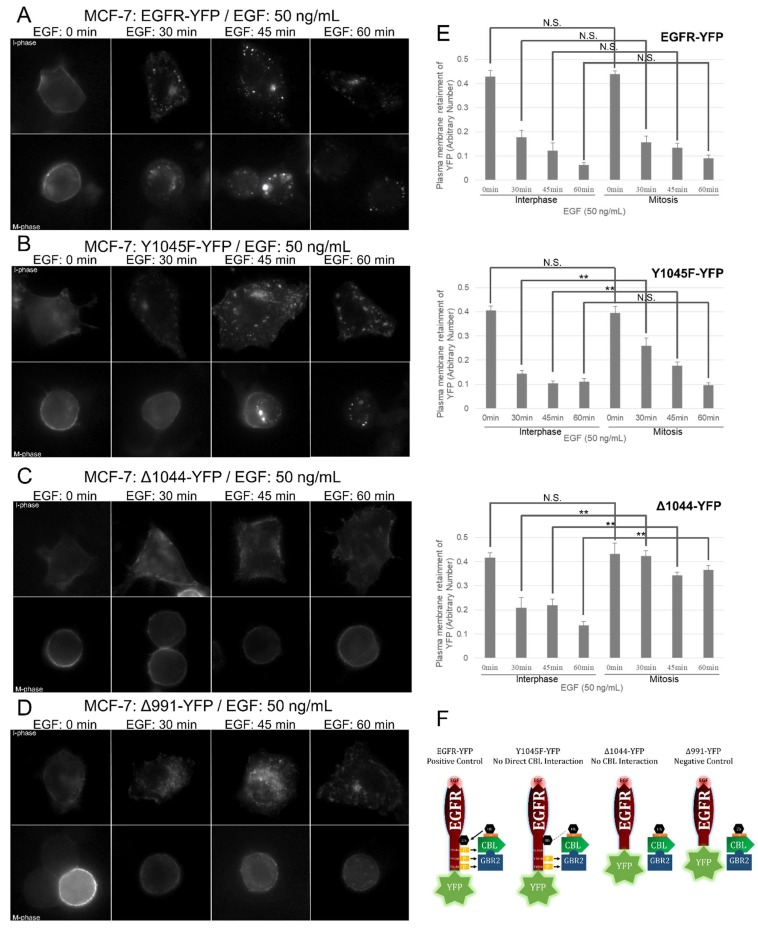
Necessity of EGFR’s CBL-binding domains for EGFR endocytosis. MCF-7 cells were transfected with (**A**) EGFR-YFP (positive control), (**B**) EGFR-Y1045F-YFP (no direct CBL binding), (**C**) EGFR-Δ1044-YFP (no CBL binding), and (**D**) EGFR-Δ991-YFP (negative control). Cells were treated with nocodazole (200 ng/mL) for 16 h and with EGF (50 ng/mL) for the specified times and observed by indirect immunofluorescence. Cell cycle phases of cells were determined by DNA morphology (not shown). (**E**) Quantification of plasma membrane retainment of YFP for (**A**–**C****)** for at least 10 cells (see Materials and Methods). The error bars are the SEMs. (**F**) Illustration of EGFR mutants used and their ability to bind CBL.

**Figure 7 cells-07-00257-f007:**
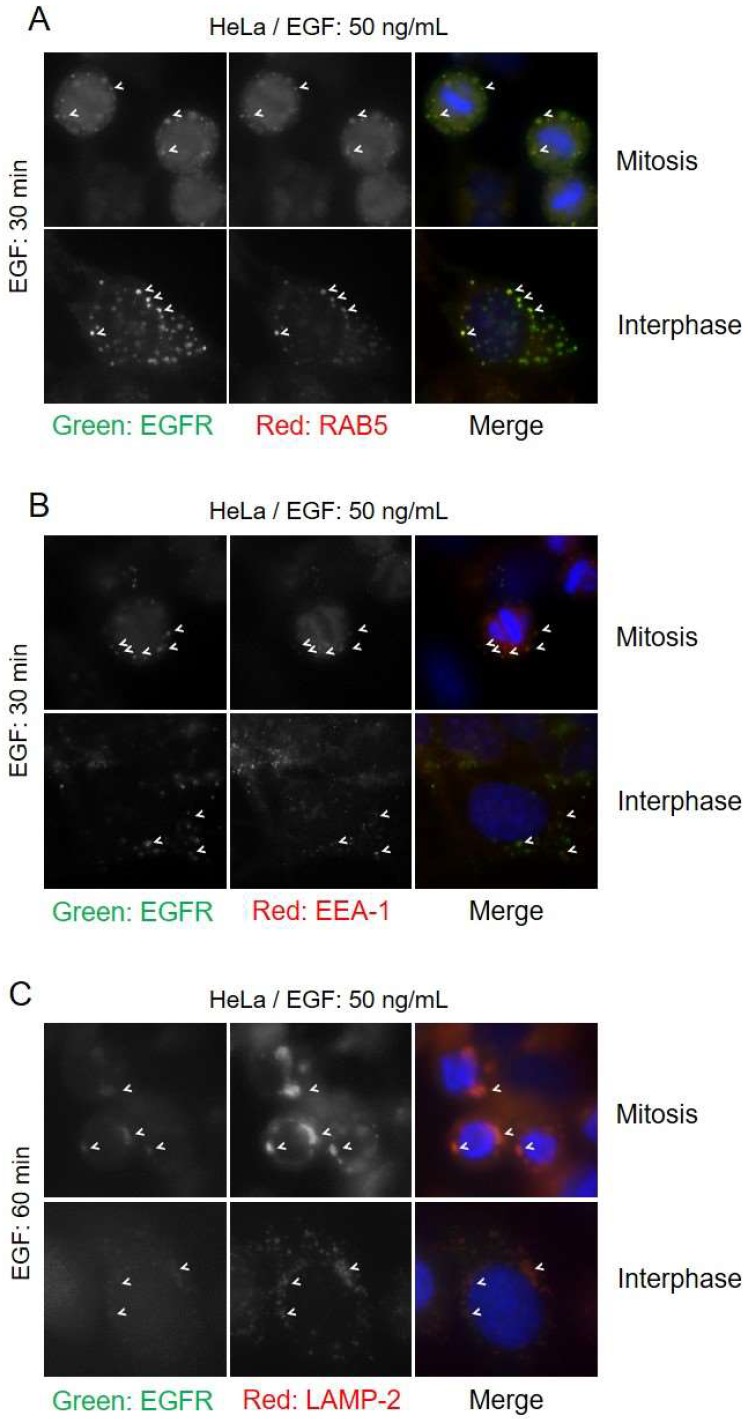
Mitotic EGFR is sorted to early endosomes and lysosomes. HeLa cells were treated with nocodazole (200 ng/mL) and EGF (50 ng/mL) for 30 min and EGFR colocalization with early endosome markers and lysosomal markers were observed by indirect immunofluorescence. Cells were stained with EGFR (green), DAPI (blue) and either: (**A**) RAB5 (red), (**B**) EEA-1 (red), or (**C**) LAMP-2 (red). Arrowheads highlight areas of colocalization.

**Figure 8 cells-07-00257-f008:**
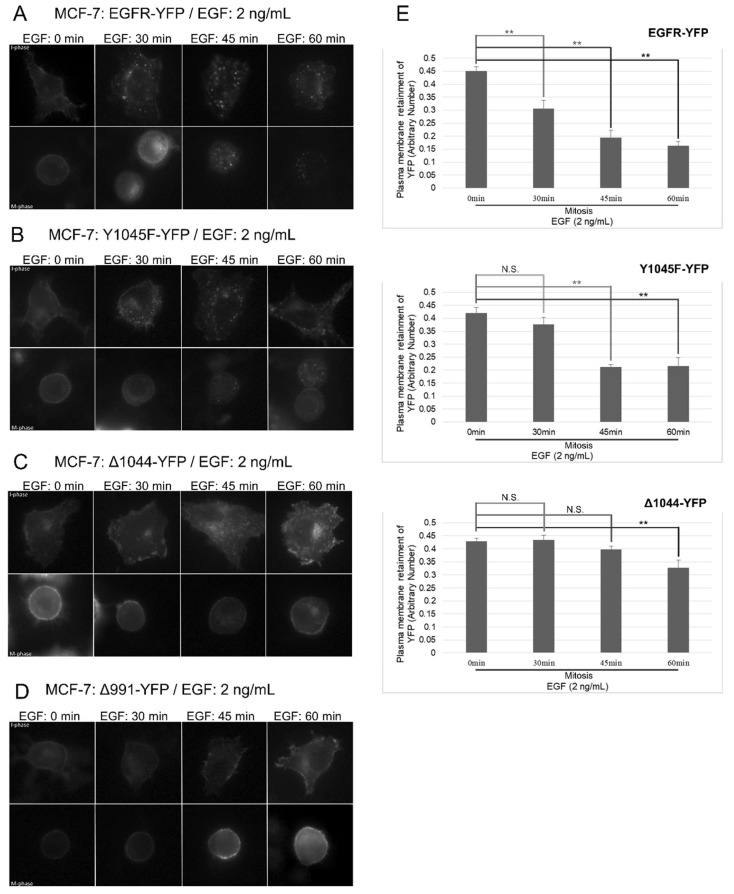
Effects of low EGF dose on mitotic EGFR endocytosis. Cells were treated with EGF (2 ng/mL) to only activate CME (clathrin-mediated endocytosis) for the specified times and observed by indirect immunofluorescence. Cell cycle phases of cells were determined by DNA morphology (not shown). MCF-7 cells were transfected with (**A**) EGFR-YFP (positive control), (**B**) EGFR-Y1045F-YFP (no direct CBL binding), (**C**) EGFR-Δ1044-YFP (no CBL binding), and (**D**) EGFR-Δ991-YFP (negative control). Cells were treated with nocodazole (200 ng/mL) for 16 h. (**E**) Quantification of plasma membrane retainment of YFP for (A–C) for at least 10 cells (see Materials and Methods). The error bars are the SEMs.

**Figure 9 cells-07-00257-f009:**
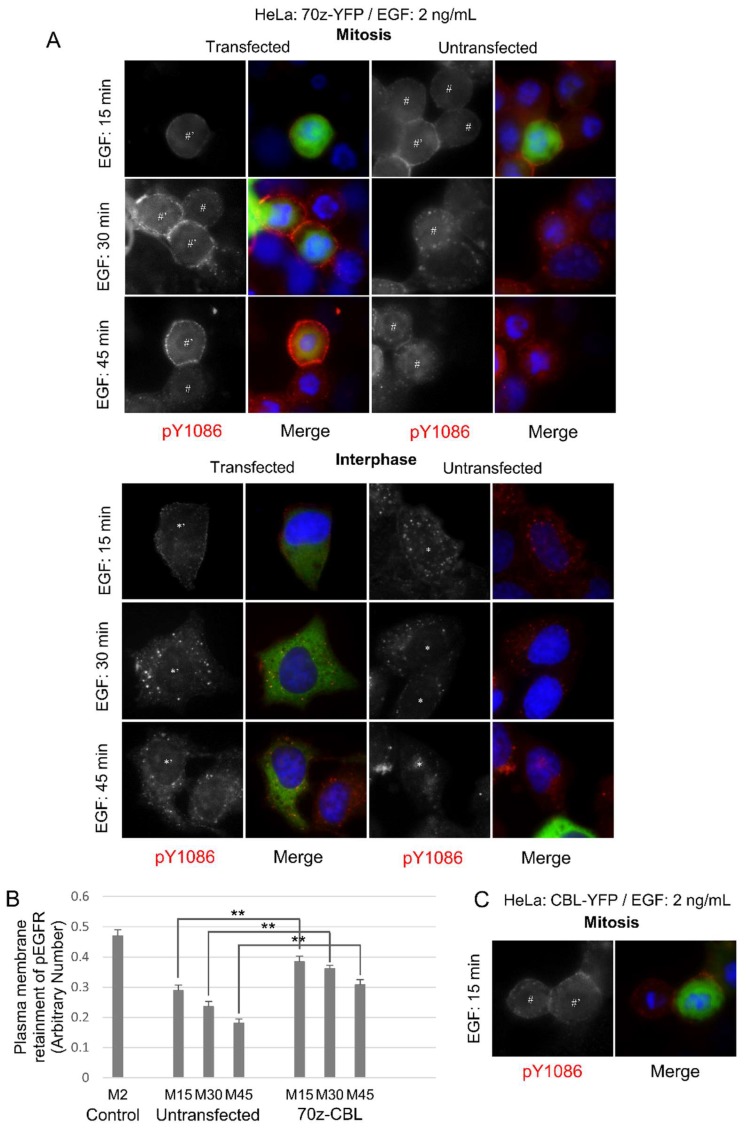
Effects of low EGF dose on endocytosis in cells with CBL alterations. (**A**) HeLa cells were transfected with dominant-negative CBL (70z-YFP) and treated with nocodazole (200 ng/mL) for 16 h. Cells were then treated with low-dose EGF (2 ng/mL) for the indicated times. Cells were stained with EGFR pY1086 (red) and DAPI (blue). * represents interphase cells, # represents mitotic cells, and ’ represents transfected cells. (**B**) Quantification of plasma membrane retainment of YFP for (**A**) for at least 10 cells (see Materials and Methods). The error bars are the SEMs. (**C**) Cells transfected with CBL-YFP were used as control.

**Figure 10 cells-07-00257-f010:**
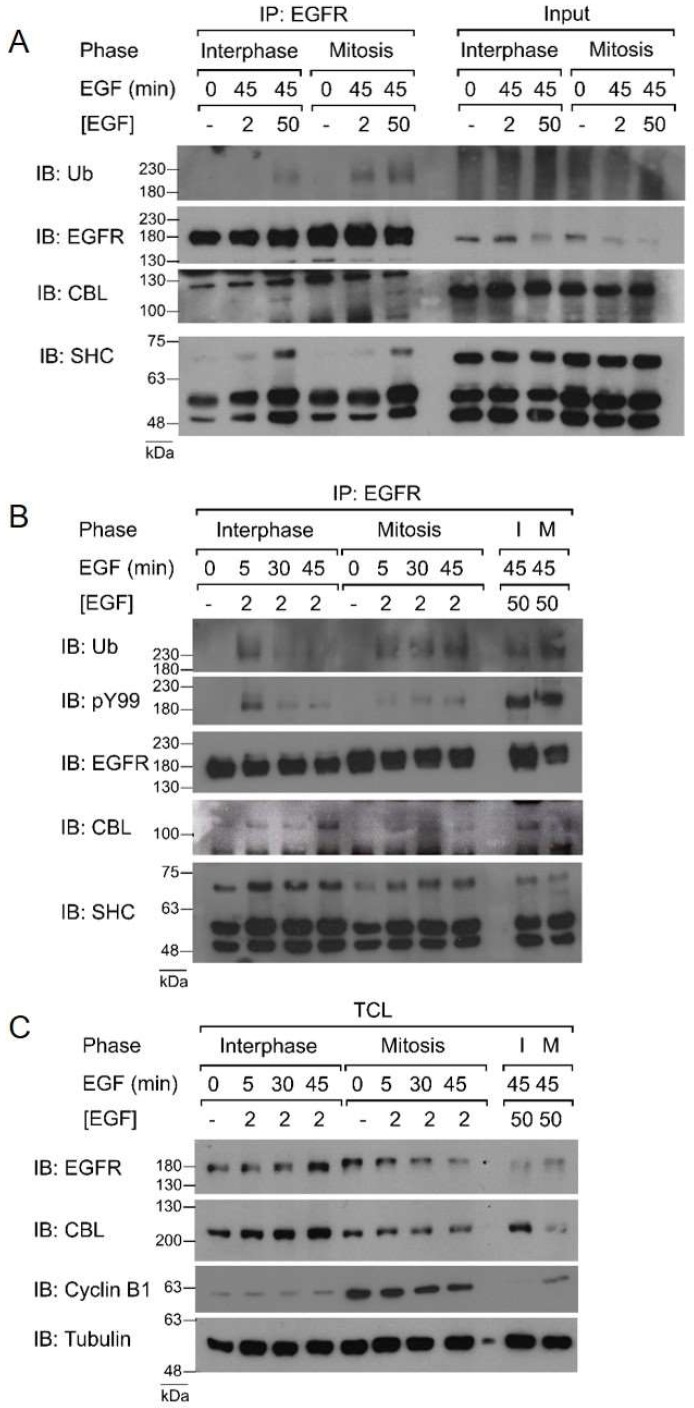
Ubiquitination and CBL-binding of EGFR with low EGF doses. Co-immunoprecipitation of EGFR from asynchronous (interphase) or nocodazole-arrested (mitosis) HeLa cells. (**A**) Cells were treated with EGF for 45 min using low- and high-dose EGF (2 and 50 ng/mL). (**B**) Cells were treated with low-dose EGF (2 ng/mL) for 0, 5, 30, or 45 min. High-dose EGF treatments for 45 min are included for reference. (**C**) Immunoblotting of TCL with the specified antibodies. Results are representative of at least two biological replicates.

**Figure 11 cells-07-00257-f011:**
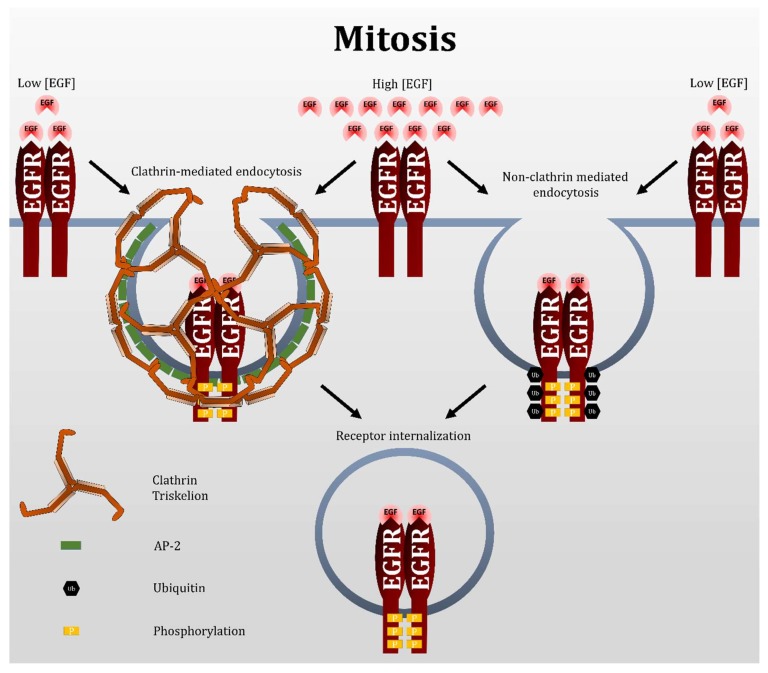
Model of EGFR endocytosis during mitosis. In interphase cells, low EGF doses (>2 ng/mL) only activate clathrin-mediated endocytosis (CME), leading to receptor recycling. High EGF doses (>20 ng/mL) also activate CME, but can also activate non-clathrin-mediated endocytosis (NCE) due to the dose-dependent activation of CBL and EGFR ubiquitination. NCE leads to lysosomal degradation of EGFR. In mitotic cells, CME is shut off. Therefore, EGFR endocytosis must proceed by NCE. Both low and high concentrations of EGF activate NCE during mitosis, and this may be because contrary to interphase cells, low EGF concentrations can activate CBL and EGFR ubiquitination. Therefore, mitotic EGFR endocytosis likely leads exclusively to lysosomal degradation.

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
