# Peer review of "Regulation of EGFR Endocytosis by CBL During Mitosis"

_cells, 2018, doi:10.3390/cells7120257_

Round 1

Reviewer 1 Report

This manuscript explores how the cell cycle effects ligand-mediated EGFR endocytosis. By comparing cells in interphase and mitosis, the authors examine the rates of endocytosis, the fates of the internalized receptors and the dependence on the E3 ubiquitin ligand c-Cbl. A strength of the manuscript is that the authors combine single cell and biochemical approaches to test for changes in endocytic trafficking under these conditions. The manuscript is well-written, the data are of high quality, and the conclusions are supported.

The overarching conclusion of the manuscript is that c-Cbl is needed for EGFR non-clathrin mediated endocytosis during mitosis.  These findings may be important in developing targeted therapies for cells with overexpressed or mutant EGFR.

Minor concerns.

1. For some of the Figure legends, there was an indication of the number of cells, but not the number of experiments. This is particularly important given that for many figures, only a single cell is shown. Further, there is no statistical analysis.

2. The section starting at 172 is formatted incorrectly.

3. In general, the manuscript is well referenced. However, from 53-55 and 172-185, there are no references and there really should be several.

4. In lines 203 and 204, the authors say that cells transfected with CBL siRNA had much less EGFR endocytosis, but there is no quantification.

5. In line 228, it would be more appropriate to say 70z-cCbl is “inhibiting CBL activity” rather than “downregulating CBL activity”.

6. Figure 6E and 8E have a bar graph with error bars, but there is no indication what those error bars are (SD? SEM?) and if the data are statistically different.

Author Response

November 27, 2018

Manuscript ID: cells-391488

Title: Regulation of EGFR endocytosis by CBL during mitosis

Response to Minor Revisions

Dear Editor,

Thank you very much to have our manuscript “Regulation of EGFR endocytosis by CBL during mitosis” reviewed. We greatly appreciate the constructive comments of the reviewer. We have revised our manuscript to incorporate the suggestions of the reviewers and addressed the concerns raised by the reviewers. We are pleased to submit the revised version of our manuscript. Our detailed response is outlined below. In addition, a marked-up copy of the changes made to the previous manuscript is also uploaded.

We hope that the modifications we have made to the manuscript will fulfill the requested revisions.

Sincerely

Zhixiang Wang, PhD

Professor

Response to Reviewer Comments

Reviewer 1 Comments

This manuscript explores how the cell cycle effects ligand-mediated EGFR endocytosis. By comparing cells in interphase and mitosis, the authors examine the rates of endocytosis, the fates of the internalized receptors and the dependence on the E3 ubiquitin ligand c-Cbl. A strength of the manuscript is that the authors combine single cell and biochemical approaches to test for changes in endocytic trafficking under these conditions. The manuscript is well-written, the data are of high quality, and the conclusions are supported.

The overarching conclusion of the manuscript is that c-Cbl is needed for EGFR non-clathrin mediated endocytosis during mitosis.  These findings may be important in developing targeted therapies for cells with overexpressed or mutant EGFR.

Minor concerns.

1. For some of the Figure legends, there was an indication of the number of cells, but not the number of experiments. This is particularly important given that for many figures, only a single cell is shown. Further, there is no statistical analysis.

Thank you for pointing this out. A section to address the number of experiments performed has been added to the Methods section (Section 2.5 Indirect immunofluorescence).

2. The section starting at 172 is formatted incorrectly.

Thanks for the keen eye. This has been corrected.

3. In general, the manuscript is well referenced. However, from 53-55 and 172-185, there are no references and there really should be several.

Good point. References have been added to these lines.

4. In lines 203 and 204, the authors say that cells transfected with CBL siRNA had much less EGFR endocytosis, but there is no quantification.

We agree with the reviewers that this is a qualitative assessment of the images.

5. In line 228, it would be more appropriate to say 70z-cCbl is “inhibiting CBL activity” rather than “downregulating CBL activity”.

Changed.

6. Figure 6E and 8E have a bar graph with error bars, but there is no indication what those error bars are (SD? SEM?) and if the data are statistically different.

This indicates SEM. This has been added to the text. Thanks for pointing this out.

Reviewer 2 Comments

It has previously been proposed that the endocytosis of transmembrane cell-surface receptors is largely shut down during mitosis. In the manuscript, however, the authors propose that EGFR endocytosis occurs mainly through CBL-mediated NCE during mitosis. Therefore, the work described in the manuscript should be published after being amended according to comments described below.

Major Comments

1)   Line 32: This sentence explaining the ligand-induced dimerization model is out of date. It is now widely accepted that the EGFR exists in a pre-formed dimeric form on the cell surface. Therefore, an alternative “rotation model” should also be mentioned here by citing a couple of references (Maruyama, I.N. Bioessays37: 959 (2015); Purba et al., Cells6:13 (2017); Maruyama, I.N. Cells3: 304 (2014); Moriki et al., J. Mol. Biol. 311: 1011 (2001)).

Thank you for the comment. The references have been included.

2)   It must be experimentally shown that EGFR-Y1045F does not interact with CBL.

Thanks for your point. We believe that EGFR-Y1045F has been studied extensively enough that this widely believed. However, we did not have the supporting references in the text and thanks to your comment, we have added them to the manuscript (Line 278).

3)   Line 313 (Fig. 7): The figure shows that EGFR endocytosis mediated by both CME and NCE take the same early and late endosomes. It does not show that EGFR endocytosed during mitosis goes exclusively to lysosomes. 

This is a good point. Unfortunately, attempts at immunofluorescence staining with markers of the recycling pathway were unsuccessful. If we were able to observe that endocytosed EGFR does not co-localize to recycling markers, we could more confidently make the claim that EGFR endocytosis during mitosis proceeds exclusively to lysosomes. However, based on the fact that: 1. CME (the major mediator of the recycling pathway) is shut off during mitosis, and 2. NCE (the major mediator of the degradative pathway) is the primary mechanism of EGFR endocytosis during mitosis, it is very likely that EGFR endocytosis during mitosis goes exclusively to lysosomes. Nevertheless, some wording in the manuscript (lines 20, 83, 399, 417) have been softened to reflect this.

4)   Line 358: Contrary to this sentence, Fig. 10A shows that CBL binds to EGFR during the interphase although the amounts are lower than those during the mitotic phase. This is consistent with ubiquitination of EGFR stimulated with 50 ug/mL. How do the authors explain this discrepancy?

Thanks for a great question. Low doses of EGF have been shown to only activate CME, while also causing CBL to bind EGFR. The CBL bound to the EGFR is also able to ubiquitinate EGFR. How is it that this doesn’t induce NCE? The answer is that with low EGF doses, the EGFR is not ubiquitinated sufficiently to undergo NCE.

A rather exquisite model to explain how CME and NCE are regulated by EGF dosage threshold has been put forward (Sigismund, S et al, 2013. Threshold-controlled ubiquitination of the egfr directs receptor fate. The EMBO journal. 32, 2140-2157). According to the experimentally validated model of Sigismund and colleagues, low doses of EGF around 1 ng/mL scarcely phosphorylate the EGFR, causing a low probability that CBL alone or CBL/GRB2 complex can bind the receptor, and as such translating into low EGFR ubiquitination. Low EGFR ubiquitination does not activate NCE, and endocytosis proceeds by CME. However, as EGF doses increase, the phosphorylation of each of the three EGFR tyrosine residues increases gradually. Then, for example, if the EGF dosage only catalyzes the phosphorylation of pY1045, and not pY1068 or pY1086, a rather unstable CBL binding occurs to the EGFR, leading to some but low ubiquitination of the EGFR. However, if all three tyrosine sites are phosphorylated, a highly stable GRB2/CBL binding can form on the EGFR, thus strongly ubiquitinating the EGFR. The kinetics involved in the probability of the CBL/GRB2 complex binding to each combination of pY site predicts a sharp rise over a narrow range of EGF dose, effectively causing EGFR ubiquitination to rise sharply once a certain dosage of EGF is applied.

We have included this discussion in the Discussion section of the revised MS.

5)   Line 373: This sentence is not correct. During the interphase, EGF at a low dose induces ubiquitination of EGFR 5 min after the stimulation. This indicates that during the interphase, EGF at a low dose induces EGFR ubiquitination, demonstrating that EGF at a low dose induces either endocytosis of EGFR via NCE, or endocytosis of EGFR via CME, in which ubiquitinated EGFR is also endocytosed via CME. The latter case is consistent with the evidence that EGFR endocytosed via CME is either recycled back to the cell surface or degraded in lysosome. This is related to the comment #4 above. Discuss these possibilities.

Thanks for your point. In addition to the comments discussed above, one important point is as below. We have shown previously that CME is inhibited during mitosis (Liu, Traffic, 2011). Thus, the ubiquitinated EGFR can only follow the NCE.

6)   It is an overstatement in the conclusions that EGFR endocytosis during mitotic phase proceeds through CBL-mediated NCE since CME activity is severely reduced during mitosis, but is not completely shut off.

As discussed in point #5, We have shown previously that CME is inhibited during mitosis (Liu, Traffic, 2011).

7)   Discuss molecular mechanisms underlying how EGF at high doses activates both CME and NCE, and how EGF at low doses activates only CME during interphase.

See point #4.

8)   Figs. 3, 6, 8, and 9: Error bars are SD or SEM? Data should be statistically analyzed, and mention what statistical analysis was used.

SEM was used. Statistical analysis has been added.

9)   There are numerous typos and grammatical errors, which must be fixed before publication. Editing by an English-native speaker is required.

Thank you for your comment and the numerous corrections in the minor comments. Ping Wee is an English-native speaker, and typos and grammatical errors were likely due to missed corrections when going back and forth with the manuscript.

Minor comments

All minor comments provided have been addressed accordingly. Most notably, higher quality figures are now provided.

Figs. 5, 6, and 8: “Interphase” and “Mitotic phage” are almost impossible to see. Replace them with larger fonts. 

Fig. 8: Very poor images. Replace them with better ones.

Throughout the manuscript: either non-transfected or un-transfected, not both.

Line 10: important role -> an important role or important roles

Line 20 and throughout the manuscript: mitotic EGFR -> EGFR during the mitotic phase

Lines 53 and 311: These sentences are misleading since CME is predominantly, but not 

exclusively, responsible for receptor recycling.

Line 104: CO2 -> CO2

Line 110: Calif -> CA

Line 120: MgCl2 -> MgCl2

Lines 120 and 131: Na3VO4 -> Na3VO4; NaN3 -> NaN3

Line 122: 16h -> 16 h

Line 180: hgher -> higher

Line 182: Most surprisingly however, -> Most surprisingly, however,

Line 188, Fig. 1 legend: Explain “TCL”.

Line 210: 45 mins -> 45-min

Line 217: Put the explanation of “*” and “#” after (A) before (B).

Lines 219-226: 70Z -> 70z

Line 228: minutes -> min

Line 232, Fig. 3B: Non-transfected <-> 70z-CBL?

Line 239: Move the explanation of “*” and “#” after (A) before (B)

Line 249: Figure 4 can be moved to supplementary since it has limited information.

Line 257: AG1418 -> AG1478

Line 267 Fig. 5C: Is this correct?

Line 269: 1 hour -> 1 h

Line 295: What are internalization motifs?

Line 326: What is TR-EGF?

Line 331: Fig. 7-8 -> Figs. 8-9?

Line 333: Y1045F-YFP -> EGFR-Y1045F-YFP

Line 334: D1044-YFP -> EGFR-D1044-YFP

Lines 337 and 338: 70z-YFP -> 70z-CBL-YFP

Line 347: In Fig. 9B, there is no data showing overexpression of CBL-YFP.

Line 376 (Fig. 9): “*” “#” “’’ cannot be distinguished. Put larger markers. In the figure legend, explain M2, M15, M30 and M45. In Fig. 9C, EGF: 15 -> EGF: 15 min

Line 397: This is an overstatement. To say “exclusively”, experimental results must be presented, which show 100% EGFR molecules are ubiquitinated. It must also been experimentally shown that CME never work during mitotic phase.

Line 549, References: Numerous typos.

Reviewer 2 Report

It has previously been proposed that the endocytosis of transmembrane cell-surface receptors is largely shut down during mitosis. In the manuscript, however, the authors propose that EGFR endocytosis occurs mainly through CBL-mediated NCE during mitosis. Therefore, the work described in the manuscript should be published after being amended according to comments described below.

Major Comments

1)   Line 32: This sentence explaining the ligand-induced dimerization model is out of date. It is now widely accepted that the EGFR exists in a pre-formed dimeric form on the cell surface. Therefore, an alternative “rotation model” should also be mentioned here by citing a couple of references (Maruyama, I.N. Bioessays37: 959 (2015); Purba et al., Cells6:13 (2017); Maruyama, I.N. Cells3: 304 (2014); Moriki et al., J. Mol. Biol311: 1011 (2001)).

2)   It must be experimentally shown that EGFR-Y1045F does not interact with CBL.

3)   Line 313 (Fig. 7): The figure shows that EGFR endocytosis mediated by both CME and NCE take the same early and late endosomes. It does not show that EGFR endocytosed during mitosis goes exclusively to lysosomes. 

4)   Line 358: Contrary to this sentence, Fig. 10A shows that CBL binds to EGFR during the interphase although the amounts are lower than those during the mitotic phase. This is consistent with ubiquitination of EGFR stimulated with 50 ug/mL. How do the authors explain this discrepancy?

5)   Line 373: This sentence is not correct. During the interphase, EGF at a low dose induces ubiquitination of EGFR 5 min after the stimulation. This indicates that during the interphase, EGF at a low dose induces EGFR ubiquitination, demonstrating that EGF at a low dose induces either endocytosis of EGFR via NCE, or endocytosis of EGFR via CME, in which ubiquitinated EGFR is also endocytosed via CME. The latter case is consistent with the evidence that EGFR endocytosed via CME is either recycled back to the cell surface or degraded in lysosome. This is related to the comment #4 above. Discuss these possibilities.

6)   It is an overstatement in the conclusions that EGFR endocytosis during mitotic phase proceeds through CBL-mediated NCE since CME activity is severely reduced during mitosis, but is not completely shut off.

7)   Discuss molecular mechanisms underlying how EGF at high doses activates both CME and NCE, and how EGF at low doses activates only CME during interphase.

8)   Figs. 3, 6, 8, and 9: Error bars are SD or SEM? Data should be statistically analyzed, and mention what statistical analysis was used.

9)   There are numerous typos and grammatical errors, which must be fixed before publication. Editing by an English-native speaker is required.

Minor comments

Figs. 5, 6, and 8: “Interphase” and “Mitotic phage” are almost impossible to see. Replace them with larger fonts. 

Fig. 8: Very poor images. Replace them with better ones.

Throughout the manuscript: either non-transfected or un-transfected, not both.

Line 10: important role -> an important role or important roles

Line 20 and throughout the manuscript: mitotic EGFR -> EGFR during the mitotic phase

Lines 53 and 311: These sentences are misleading since CME is predominantly, but not 

exclusively, responsible for receptor recycling.

Line 104: CO2 -> CO2

Line 110: Calif -> CA

Line 120: MgCl2 -> MgCl2

Lines 120 and 131: Na3VO4 -> Na3VO4; NaN3 -> NaN3

Line 122: 16h -> 16 h

Line 180: hgher -> higher

Line 182: Most surprisingly however, -> Most surprisingly, however,

Line 188, Fig. 1 legend: Explain “TCL”.

Line 210: 45 mins -> 45-min

Line 217: Put the explanation of “*” and “#” after (A) before (B).

Lines 219-226: 70Z -> 70z

Line 228: minutes -> min

Line 232, Fig. 3B: Non-transfected <-> 70z-CBL?

Line 239: Move the explanation of “*” and “#” after (A) before (B)

Line 249: Figure 4 can be moved to supplementary since it has limited information.

Line 257: AG1418 -> AG1478

Line 267 Fig. 5C: Is this correct?

Line 269: 1 hour -> 1 h

Line 295: What are internalization motifs?

Line 326: What is TR-EGF?

Line 331: Fig. 7-8 -> Figs. 8-9?

Line 333: Y1045F-YFP -> EGFR-Y1045F-YFP

Line 334: D1044-YFP -> EGFR-D1044-YFP

Lines 337 and 338: 70z-YFP -> 70z-CBL-YFP

Line 347: In Fig. 9B, there is no data showing overexpression of CBL-YFP.

Line 376 (Fig. 9): “*” “#” “’’ cannot be distinguished. Put larger markers. In the figure legend, explain M2, M15, M30 and M45. In Fig. 9C, EGF: 15 -> EGF: 15 min

Line 397: This is an overstatement. To say “exclusively”, experimental results must be presented, which show 100% EGFR molecules are ubiquitinated. It must also been experimentally shown that CME never work during mitotic phase.

Line 549, References: Numerous typos.

Author Response

(The authors gave the same response as above.)
